# Factors in COVID-19 vaccine uptake in five racial/ethnic Colorado communities: A report from the Colorado CEAL project

Sarah E. Brewer[1,2]*, Kaitlyn B. Bertin[1], Krithika Suresh[3], Crystal LoudHawk-Hedgepeth[4,5], Montelle Tamez[4], Jenna E. Reno[6], Bethany M. Kwan[1,7], Donald E. Nease[1,2,4]

1 Adult and Child Center for Outcomes Research and Delivery Science, University of Colorado Anschutz Medical Campus, Aurora, CO, United States of America, 2 Department of Family Medicine, University of Colorado School of Medicine, University of Colorado Anschutz Medical Campus, Aurora, CO, United States of America, 3 Department of Biostatistics and Informatics, Colorado School of Public Health, University of Colorado Anschutz Medical Campus, Aurora, CO, United States of America, 4 Colorado Clinical and Translational Sciences Institute, University of Colorado Anschutz Medical Campus, Aurora, CO, United States of America, 5 American Indian College Fund, Denver, CO, United States of America, 6 Center for Communication and Engagement Research, RTI International, Aurora, CO, United States of America, 7 Department of Emergency Medicine, University of Colorado School of Medicine, University of Colorado Anschutz Medical Campus, Aurora, CO, United States of America

* Sarah.Brewer@CUAnschutz.edu

## Abstract

### Purpose

To understand motivators, concerns, and factors associated with COVID-19 vaccine initiation for adults in five racial/ethnic communities across Colorado.

### Methods

Community-based data collectors surveyed participants from five Colorado communities (urban and rural Latina/o/x, urban Black, rural African American immigrant, and urban American Indian) about vaccine attitudes, intentions, and uptake from September to December 2021. Bivariate and multivariable logistic regression models were used to examine factors associated with the primary outcome of COVID-19 vaccine "initiation."

### Results

Most participants (71.1%) reported having initiated COVID-19 vaccination; vaccine series completion was 65.1%. Both motivators and concerns about COVID-19 vaccines were prevalent. Vaccine hesitancy (OR: 0.41, 95% CI:0.32–0.53; p < .001) and low perceptions of COVID-19 vaccination social norms (OR: 0.48, 95% CI:0.27–0.84; p = .01) were associated with vaccine initiation.

### Conclusion

Despite the limitation of a moderate sample size, our findings support the need for further interventions to increase vaccination against COVID-19 by reducing vaccine hesitancy and improving perceived social norms of vaccination in underserved Colorado communities.

**Data Availability Statement:** Data cannot be shared publicly because Common Survey data from NIH-funded CEAL research are only shared

through the permission-based online portal and cannot be shared via email. Researchers requesting access to the data must adhere to the data use policy and the embargo period. Data are available from the CEAL portal (contact via CETAC@westat.com) for researchers who meet the criteria for access to confidential data. The data underlying the results presented in the study are available from the Common Survey data portal managed my Westat - contact via CETAC@westat.com.

**Funding:** Research reported in this publication was supported by the National Institutes of Health under the Sub-OTA No. 6793-02-S018. The content is solely the responsibility of the authors and does not necessarily represent the official views of the National Institutes of Health. Study data were collected and managed using REDCap electronic data capture tools hosted at the University of Colorado. REDCap (Research Electronic Data Capture) is a secure, web-based application designed to support data capture for research studies, providing: 1) an intuitive interface for validated data entry; 2) audit trails for tracking data manipulation and export procedures; 3) automated export procedures for seamless data downloads to common statistical packages; and 4) procedures for importing data from external sources. This project and related publication were supported by NIH/NCATS Colorado CTSA Grant Number UL1 TR002535. Its contents are the authors' sole responsibility and do not necessarily represent official NIH views." The funders had no role in study design, data collection and analysis, decision to publish, or preparation of the manuscript.

**Competing interests:** I have read the journal's policy and the authors of this manuscript have the following competing interests: Sarah Brewer served on an advisory board for Merck. The company did not play any role in this research, and we do not receive research funding from them. All other authors have no conflicts to declare. This does not alter our adherence to PLOS ONE policies on sharing data and materials.

## Implications

To improve trust in vaccines and promote vaccine uptake, community messaging should be tailored to vaccination motivators and concerns and demonstrate COVID-19 vaccination as the community default.

## Introduction

The COVID-19 pandemic has had disproportionate impacts on certain communities nationally and in Colorado. As of January 2023, the US accounts for over 102 million cases and 1.1 million deaths, representing a disproportionate burden of disease and mortality relative to other nations [1]. Colorado had over 1.7 million cases and 14,500 deaths due to COVID-19 in 2020–2022. In Colorado, COVID-19 impacts on racial/ethnic minority and rural communities reveal striking disparities. In early 2021, Latino/a/x communities had experienced case rates of COVID-19 at 27.7% compared with 21.7% in the overall population. Among American Indian/Alaska Natives (AI/AN), the death rate due to COVID-19 was 1.4 times higher than their proportion of the population. Geographically, the top ten counties in Colorado for COVID-19 incidence in early 2021 were rural, with the worst ranked county experiencing a case rate of over 32,000 per 100,000 [2]. Death rates show similar disparities for rural area, with some rural counties experiencing death rates over three times those for Denver County [2]. These disproportionate impacts of COVID-19 on certain populations have been associated with underlying socio-economic factors, but these were also exacerbated by misinformation about prevention measures, testing, and vaccines [3–5].

In addition to well-documented state and national disparities in morbidity and mortality from COVID-19, these communities also lagged in vaccination rates. In Colorado in early 2021, persons who identified as Hispanic had received only 5.0% of all vaccines while representing 21.7% of the population; Blacks/African Americans (Black/AA) had received 2.1% of the delivered vaccines while representing 3.92% of the population. These disparities have persisted over time. As of December 31, 2022 in Colorado, only 42% of Hispanic, 69% of Black/AA, and 74.5% of AI/AN people had received at least one dose of COVID-19 vaccine, while 78% of the total population and 81% of whites had received a COVID-19 vaccine [6]. Additionally, 75% of the Coloradoans who had received at least one booster dose were white. Vaccination disparities have not been limited to racial/ethnic communities. State surveillance data show that 71% of Coloradans have completed a vaccine series, but 20 rural counties in Colorado have completion rates for the primary series below 60% and all of Colorado's urban counties have rates above 60% [6]. National panel surveys have shown lower rates of uptake for low-income and non-Hispanic Black adults [7] and Hispanic adults [8]. Additionally, vaccine hesitancy was higher among both Black/AA (42%) and Hispanic (30%) populations than among non-Hispanic whites (26%) [9, 10].

Numerous factors, including hesitancy, medical mistrust, and perceived social norms of vaccination have been associated with acceptance of COVID-19 and other vaccines. Since the COVID-19 pandemic began, hesitancy about potential and then real vaccines has been prevalent [11, 12]. Concerns have included both real and speculative questions regarding its effectiveness, side effects, short- and long-term impacts on fertility and health, and the speed of development [11, 13]. While many of these concerns echo the concerns voiced by those with hesitancy toward other vaccines, throughout the pandemic, estimates of vaccine hesitancy have been higher among Black/AA, Latino/a/x, and rural populations [13–16]. Medical mistrust–the perception that healthcare organizations and medical personnel do not treat all

patients fairly or are untrustworthy–has also been associated with lower uptake of COVID-19 and other vaccines [17, 18]. Finally, social norms–or the perception that vaccination is an expected or normal behavior among people like oneself–have long been associated with vaccine acceptance and this relationship appears to hold true for recent COVID-19 vaccines [19, 20]; however, little evidence exists to describe these factors specifically in racial and ethnic minority communities with lower vaccine uptake. Understanding the impact of concerns, motivators, mistrust and social norms on COVID-19 vaccine uptake in these underserved communities could help inform interventions to promote vaccination uptake.

As part of the National Institutes of Health (NIH) and National Heart, Lung, and Blood Institute's (NHLBI) Community Engagement Alliance (CEAL) Against COVID-19 Disparities initiative, [21] the Colorado CEAL (CO-CEAL) team sought to address disparities in the impacts of COVID-19 disease and vaccine uptake starting in spring 2021. The CO-CEAL program aimed to assess concerns and misinformation about COVID-19 vaccines to inform efforts to promote vaccine uptake in five regional racial/ethnic communities. Specifically, we sought to understand (a) vaccine initiation and completion among adults and children (ages 12–17) (the vaccine was not available for those 11 and under at the time of data collection); (b) motivators for, concerns regarding, and intention toward COVID-19 vaccination; and (c) factors including mistrust social norms associated with adult vaccine initiation across five disproportionately impacted racial/ethnic communities in Colorado.

## Methods

This paper reports on baseline survey data collected within the context of a COVID-19 vaccine promotion pre-post interventional study in the fall of 2021. The study was designated as exempt by the Colorado Multiple Institutional Review Board (COMIRB). Despite that, participants provided electronic written informed consent in the REDCap survey (online, in the REDCap app, or on paper copies of the survey) before participation, with assistance from a member of the study team as needed. Limited identifying information was collected to contact participants for future waves of data collection and identifiers were stored on the secure RED-Cap servers and separated from the analytic dataset. Data access was limited to the analytic team to ensure participant privacy and data security.

### Setting & participants

Survey participants were recruited from five Colorado regional communities: urban Latino/a/x, rural Latino/a/x, urban Black/AA, rural African immigrant (primarily from Somalia), and urban American Indian/Alaska Native (AI/AN). These community groups were selected for the disparities they each experienced during the COVID-19 pandemic and priorities expressed by the National Institutes of Health. As the goal was to assess community-wide attitudes and concerns about vaccination, in each regional community, eligibility criteria were broad. Participants were eligible if they were Colorado residents and lived in one of the included regional communities, identified with one of the designated racial or ethnic groups of that community, and were at least 18 years of age.

### Recruitment & data collection

Survey participants were recruited using a Community Connector model adapted from other community-engaged studies [22–24]. Each community had a Community Connector (CC) who liaised between University staff and community-based collection teams and were trained members from the community of focus. Data collectors were trained in human subjects protections using COMIRB-designed training for community members, recruitment and consent

procedures, and survey administration. Each data collector undertook a combined convenience, venue-based, and snowball sampling approach to recruit and enroll 200 members per community, for a target total sample size of 1000 that was powered to detect a clinically meaningful difference in vaccination uptake across communities post-intervention.

Data collection took place from September to December 2021. Surveys were available in two formats (paper or an online REDCap survey) and three languages (English, Spanish, Somali). Survey administration ranged from self-completed to fully-facilitated by the data collector depending on participant needs and preferences. Data collectors used either remote or in-person administration strategies based on local COVID-19 incidence.

## Measures

The survey instrument incorporated the NIH CEAL Common Survey 1 (available at https://cctsi.cuanschutz.edu/community/co-ceal) as well as additional domains based on the CO-CEAL interests [25]. The CEAL Common Survey includes a variety of established valid and reliable measures of social determinants of health, healthcare use and access, health status, trust of public health information sources, COVID-19 prevention behaviors, vaccination acceptance (including receipt, attitudes, motivators, concerns, intentions for vaccination and boosters), vaccine acceptance for one's children, and knowledge and attitudes about COVID-19-related clinical trials.

**Primary outcome.** The primary outcome measure was initiation of the COVID-19 vaccination series (at least one dose), which was captured by the item "Have you received a COVID-19 vaccine?" and dichotomized as Yes (1) vs. No (0).

**Factor variables.** From the CEAL Common Survey we adapted two questions to ascertain reasons for vaccinating ('motivators') and reasons not to vaccinate ('concerns'). The CO-CEAL survey included additional domains for vaccine hesitancy, medical mistrust, social norms, influenza vaccine uptake, impacts of the pandemic on social needs and healthcare, trusted information sources, preferred social media platform, and clinical trials (not all are discussed in the present study). General vaccine hesitancy was measured using a 4-item scale adapted from the Vaccine Attitudes Examination (VAX) Scale [26]. A total score ranging from 1 to 6, with higher scores indicating greater vaccine hesitancy, was calculated as the mean of individual item scores (on a 6-point Likert scale) if at least two items were answered. Medical mistrust was measured using a 4-item scale adapted from the Group-Based Medical Mistrust Scale (GBMMS) [27]. Our adaptations of the VAX and GBMMS were conducted through conversation with a small group of CCs and other advisors from each community and included changes to items to specifically capture mistrust in the context of COVID-19 pandemic and to focus on perceptions of the medical system as trustworthy, rather than on individuals' trust in the system (e.g., "When it comes to COVID-19, people from my racial/ethnic group will receive the same medical care from healthcare providers as people from other groups.") [27]. A total score ranging from 1 to 5, with higher scores indicating greater group-based medical mistrust regarding COVID-19, was calculated as the mean of individual item scores (on a 5-point Likert scale) if at least two items were answered. A composite measure of social norms was derived by averaging the scores of two items (with 5-level responses scored 1 to 5) adapted from prior work on vaccine behaviors [28–31] and designed to capture perceptions of both descriptive group norms ("Thinking about people of your own race/ethnicity, how many of them do you think will get the COVID-19 vaccine?") and subjective norms ("Of the people close to you, what proportion of them would want you to get the COVID-19 vaccine?"), and creating a three-level categorical measure (low, medium, high) for the perception of social norms for COVID-19 vaccination.

We assessed completion of a COVID-19 vaccination series, vaccination intention for the primary vaccine series among those who were not yet vaccinated, intention to receive the COVID-19 booster dose among those who completed the primary vaccine series, initiation and completion of the COVID-19 vaccination series for children ages 12–17 living in their household, and intention to get a COVID-19 vaccine for children ages 0–11 living in their household when available. We also measured health literacy with a single item from the CEAL Common Survey ("How often do you need someone to help you read written information from your doctor or drug store?") scored on a 5-point scale ranging from 0 ("never") to 4 ("always").

**Demographics.** The CEAL Common Survey included demographic questions including insurance, income, gender, age, education, and immigration status.

## Data analysis

Descriptive statistics were calculated for the survey responses overall and by community. Logistic regression was used to assess associations between the primary outcome of vaccination initiation and potential factors. To account for potential dependence in the data among survey respondents recruited by the same community data collector, a generalized estimating equation approach was used with an independence working correlation structure selected based on quasi-information criterion (QIC) [32]. For categorical variables, the largest group was selected as the reference group (except for health insurance coverage, for which "private" was selected to better reveal group differences). The final multivariable model was adjusted for demographic variables identified a priori (age, gender, education, income, community) based on observed factors associated with vaccine uptake during the early roll-out of the COVID-19 vaccines, and any predictors that were identified to be associated ($p<0.2$) with the outcome in bivariate analyses. Additionally, we adhered to the principle of limiting to 10 events per parameter; thus, with and effective sample size of n = 235 in our non-uptake group, we had a goal to limit the total number of model parameters to 24 or less. Odds ratios (OR), 95% confidence intervals (CI), and p-values are reported, and statistical significance is assessed at the 0.05 level. All analysis was conducted using SAS version 9.4 (SAS Institute Inc).

## Results

Across five communities, 841 participants responded to the survey; 29 (3%) were excluded due to incomplete responses to the primary outcome measure and the final analytic sample was 812 participants. Community representation was 23.3% urban Latino/a/x, 24.0% rural Latino/a/x, 20.8% urban Black/AA, 24.0% rural Black/AA, and 7.9% urban AI/AN. Participant demographics overall and by community are presented in **Table 1**. The overall sample had an average age of 42.2 years (SD = 16.8) and included 57.3% women. Of respondents, 32.6% were born outside the US, 60.7% had a high school education or less, and 60.7% reported a household income of less than $50,000. Of all surveys, 88.2% were completed in English, 6.2% in Spanish, and 5.7% in Somali.

### COVID-19 vaccine uptake and intention

Rates of vaccination series initiation and completion, and initiation and booster intentions are shown in **Table 2**. Of participants, 71.1% (n = 577) reported having received ≥1 dose of a COVID-19 vaccine; series completion was 65.1% (n = 529). There were differences across communities for both initiation (62.1% urban Black/AA, 93.8% AI/AN) and completion (52.7% urban Black/AA, 89.1% AI/AN). Among those who completed the primary series, 69.6% intended to receive a booster dose (56.2% urban Black/AA, 82.0% urban Latino/a/x). Of

**Table 1. Sample characteristics at baseline, overall and by community cohort, Colorado, 2021.**

| Characteristic | CO-CEAL Sample Overall (N = 812) | Urban Latino/a/x (N = 189) | Rural Latino/a/x (N = 195) | Urban Black/AA (N = 169) | Rural Black/AA (N = 195) | Urban AI/AN (N = 64) |
|---|---|---|---|---|---|---|
| **Survey language** | | | | | | |
| English | 716 (88.2%) | 170 (89.9%) | 164 (84.1%) | 169 (100%) | 149 (76.4%) | 64 (100%) |
| Spanish | 50 (6.2%) | 19 (10.1%) | 31 (15.9%) | 0 (0.0%) | 0 (0.0%) | 0 (0.0%) |
| Somali | 46 (5.7%) | 0 (0.0%) | 0 (0.0%) | 0 (0.0%) | 46 (23.6%) | 0 (0.0%) |
| **Age (years)** | 42.2 (16.8) | 48.7 (16.2) | 42.1 (17.5) | 44.3 (18.1) | 35.5 (13.2) | 37.7 (14.5) |
| **Gender** | | | | | | |
| Man | 338 (41.6%) | 59 (31.2%) | 63 (32.3%) | 78 (46.2%) | 110 (56.4%) | 28 (43.8%) |
| Woman | 465 (57.3%) | 129 (68.3%) | 129 (66.2%) | 89 (52.7%) | 84 (43.1%) | 34 (53.1%) |
| Other | 5 (0.6%) | 0 (0.0%) | 2 (1.0%) | 1 (0.6%) | 1 (0.5%) | 1 (1.6%) |
| **Immigrant** | | | | | | |
| Yes | 265 (32.6%) | 36 (19.0%) | 32 (16.4%) | 10 (5.9%) | 187 (95.9%) | 0 (0.0%) |
| No | 529 (65.1%) | 150 (79.4%) | 159 (81.5%) | 150 (88.8%) | 7 (3.6%) | 63 (98.4%) |
| **Education** | | | | | | |
| High school/GED or less | 493 (60.7%) | 118 (62.4%) | 125 (64.1%) | 110 (65.1%) | 108 (55.4%) | 32 (50.0%) |
| Associate's/tech. degree | 116 (14.3%) | 31 (16.4%) | 31 (15.9%) | 25 (14.8%) | 13 (6.7%) | 16 (25.0%) |
| Bachelor's/grad. degree | 117 (14.4%) | 35 (18.5%) | 34 (17.4%) | 29 (17.2%) | 4 (2.1%) | 15 (23.4%) |
| Prefer no answer/missing | 86 (10.6%) | 5 (2.6%) | 5 (2.6%) | 5 (3.0%) | 70 (35.9%) | 1 (1.6%) |
| **Household income** | | | | | | |
| Less than $25,000 | 300 (36.9%) | 58 (30.7%) | 70 (35.9%) | 77 (45.6%) | 78 (40.0%) | 17 (26.6%) |
| $25,000-$49,999 | 193 (23.8%) | 43 (22.8%) | 50 (25.6%) | 34 (20.1%) | 48 (24.6%) | 18 (28.1%) |
| $50,000-$74,999 | 87 (10.7%) | 22 (11.6%) | 21 (10.8%) | 13 (7.7%) | 17 (8.7%) | 14 (21.9%) |
| $75,000 or more | 72 (8.9%) | 31 (16.4%) | 12 (6.2%) | 17 (10.1%) | 3 (1.5%) | 9 (14.1%) |
| Prefer no answer/missing | 160 (19.7%) | 35 (18.5%) | 42 (21.5%) | 28 (16.6%) | 49 (25.1%) | 6 (9.4%) |
| **Has usual place for care** | | | | | | |
| Yes | 702 (86.5%) | 158 (83.6%) | 177 (90.8%) | 129 (76.3%) | 185 (94.9%) | 53 (82.8%) |
| No | 107 (13.2%) | 30 (15.9%) | 18 (9.2%) | 38 (22.5%) | 10 (5.1%) | 11 (17.2%) |
| **Health insurance** | | | | | | |
| None/don't know | 163 (20.1%) | 39 (20.6%) | 43 (22.1%) | 33 (19.5%) | 39 (20.0%) | 9 (14.1%) |
| Public | 384 (47.3%) | 86 (45.5%) | 88 (45.1%) | 100 (59.2%) | 88 (45.1%) | 22 (34.4%) |
| Private | 263 (32.4%) | 62 (32.8%) | 64 (32.8%) | 36 (21.3%) | 68 (34.9%) | 33 (51.6%) |
| **Health literacy score[a]** | 0.8 (1.3) | 0.5 (1.1) | 0.4 (0.9) | 0.4 (0.8) | 1.8 (1.7) | 0.3 (0.5) |
| **Vaccination hesitancy, total score[b]** | 3.3 (1.0) | 3.3 (1.1) | 3.4 (1.0) | 3.5 (0.9) | 3.1 (1.0) | 3.1 (0.8) |
| **Medical mistrust, total score[c]** | 2.9 (0.8) | 3.0 (0.9) | 2.8 (0.8) | 3.2 (0.7) | 2.5 (0.8) | 2.9 (0.6) |
| **COVID-19 vaccination social norms composite[d]** | | | | | | |
| Low | 254 (31.3%) | 37 (19.6%) | 64 (32.8%) | 77 (45.6%) | 69 (35.4%) | 7 (10.9%) |
| Medium | 213 (26.2%) | 52 (27.5%) | 64 (32.8%) | 51 (30.2%) | 24 (12.3%) | 22 (34.4%) |
| High | 337 (41.5%) | 98 (51.9%) | 65 (33.3%) | 37 (21.9%) | 102 (52.3%) | 35 (54.7%) |

AA = African American; AI/AN = American Indian/Alaska Native; CO-CEAL = Colorado CEAL; SD = standard deviation.

Note: Descriptive statistics are presented as either "n (%)" or "mean (SD)". Percentages may not add to 100% within the column for variables with missing data. Missing or "prefer not to answer" responses are presented as a category if representing at least 5% of the total sample.

[a] Range = 0 to 4; higher scores indicate more frequent need for help reading written information from doctor or drug store.

[b] Based on modifaied Vaccination Attitudes (VAX) Scale; range = 1 to 6; higher scores indicate greater vaccine hesitancy.

[c] Based on modified Group-Based Medical Mistrust Scale (GBMMS); range = 1 to 5; higher scores indicate greater group-based medical mistrust.

[d] Classified based on total score (range = 1 to 5) calculated as mean of 2 items (subjective and descriptive norms), with "low" = 1 to 2.5, "medium" = 3 to 3.5, "high" = 4 to 5. Higher scores reflect greater perceived social norms regarding COVID-19 vaccination (perceiving a larger percentage of people who would get or want the respondent to get a vaccine).

**Table 2. COVID-19 vaccination outcomes at baseline, overall and by community cohort, Colorado, 2021.**

| Characteristic | CO-CEAL Sample Overall (N = 812) | Urban Latino/a/x (N = 189) | Rural Latino/a/x (N = 195) | Urban Black/AA (N = 169) | Rural Black/AA (N = 195) | Urban AI/AN (N = 64) |
|---|---|---|---|---|---|---|
| **Initiated COVID-19 vaccine (received any dose)** | | | | | | |
| Yes | 577 (71.1%) | 140 (74.1%) | 144 (73.8%) | 105 (62.1%) | 128 (65.6%) | 60 (93.8%) |
| No | 235 (28.9%) | 49 (25.9%) | 51 (26.2%) | 64 (37.9%) | 67 (34.4%) | 4 (6.3%) |
| **Completed COVID-19 vaccine course** | | | | | | |
| Yes | 529 (65.1%) | 133 (70.4%) | 136 (69.7%) | 89 (52.7%) | 114 (58.5%) | 57 (89.1%) |
| No | 271 (33.4%) | 52 (27.5%) | 57 (29.2%) | 78 (46.2%) | 79 (40.5%) | 5 (7.8%) |
| **Likely on 1–7 scale to get COVID-19 booster (if completed course)** | **N = 529** | **N = 133** | **N = 136** | **N = 89** | **N = 114** | **N = 57** |
| Yes | 368 (69.6%) | 109 (82.0%) | 89 (65.4%) | 50 (56.2%) | 76 (66.7%) | 44 (77.2%) |
| No | 154 (29.1%) | 20 (15.0%) | 45 (33.1%) | 39 (43.8%) | 38 (33.3%) | 12 (21.1%) |
| **Likely on 1–7 scale to get COVID-19 vaccine in the next few months (if not yet done)** | **N = 235** | **N = 49** | **N = 51** | **N = 64** | **N = 67** | **N = 4** |
| Yes | 71 (30.2%) | 14 (28.6%) | 9 (17.6%) | 11 (17.2%) | 37 (55.2%) | 0 (0.0%) |
| No | 159 (67.7%) | 34 (69.4%) | 39 (76.5%) | 52 (81.3%) | 30 (44.8%) | 4 (100%) |
| **Child ages 12–17 initiated COVID-19 vaccine (received any dose)** | **N = 195** | **N = 42** | **N = 63** | **N = 36** | **N = 34** | **N = 20** |
| Yes | 67 (34.4%) | 14 (33.3%) | 27 (42.9%) | 7 (19.4%) | 7 (20.6%) | 12 (60.0%) |
| No | 76 (39.0%) | 19 (45.2%) | 18 (28.6%) | 18 (50.0%) | 20 (58.8%) | 1 (5.0%) |
| **Child ages 12–17 completed COVID-19 vaccine course** | | | | | | |
| Yes | 55 (28.2%) | 13 (31.0%) | 22 (34.9%) | 5 (13.9%) | 3 (8.8%) | 12 (60.0%) |
| No | 86 (44.1%) | 20 (47.6%) | 22 (34.9%) | 19 (52.8%) | 24 (70.6%) | 1 (5.0%) |
| **Likely on 1–7 scale to get child ages 0–11 COVID-19 vaccine when available[a]** | **N = 240** | **N = 58** | **N = 67** | **N = 46** | **N = 51** | **N = 18** |
| Yes | 102 (42.5%) | 21 (36.2%) | 27 (40.3%) | 14 (30.4%) | 30 (58.8%) | 10 (55.6%) |
| No | 118 (49.2%) | 31 (53.4%) | 31 (46.3%) | 30 (65.2%) | 21 (41.2%) | 5 (27.8%) |

AA = African American; AI/AN = American Indian/Alaska Native.

Note: Descriptive statistics are presented as "n (%)". Percentages may not add to 100% within the column for variables with missing data. Missing data represented <5% of the total sample for all presented variables. Denominators for child vaccine outcomes represent the number of participants who reported children living in their household within the given age range.

[a] Most data were collected prior to approval of a COVID-19 vaccine for ages 5–11 in the US.

those who had not yet initiated COVID-19 vaccination (n = 235), 30.2% reported they intended to receive a dose in the coming months. Of respondents with children ages 12–17 (n = 195), 34.4% reported the child had received a COVID-19 vaccine. Of respondents with children ages 0–11 (n = 240), 42.5% reported positive intentions to vaccinate those children.

## Motivators and barriers to vaccination

Participant-reported motivators and concerns for receiving a COVID-19 vaccine are shown in Table 3 for both the overall sample and stratified by vaccine initiation status. The most common motivators for vaccination were to keep one's family (69.8%), self (58.1%), and community (50.6%) safe, with higher reporting of safety motivators among those who had vaccine initiation compared with those who did not. Not wanting to get sick from COVID-19 was ranked fifth and reported by 32.6% of respondents. Seventy percent of respondents reported at least one concern about COVID-19 vaccines. Among those who had not initiated vaccination

**Table 3. Reasons for (motivators) and against (concerns) getting a COVID-19 vaccine, overall and by initiation status, Colorado, 2021.**

| Reason for/against vaccination | Overall (N = 812) | | Initiated COVID-19 vaccine (N = 577) | | Did not initiate COVID-19 vaccine (N = 235) | |
|---|---|---|---|---|---|---|
| | Rank | n (%) | Rank | n (%) | Rank | n (%) |
| *Motivators* | | | | | | |
| I want to keep my family safe | 1 | 567 (69.8%) | 1 | 471 (81.6%) | 1 | 96 (40.9%) |
| I want to keep myself safe | 2 | 472 (58.1%) | 2 | 402 (69.7%) | 3 | 70 (29.8%) |
| I want to keep my community safe | 3 | 411 (50.6%) | 3 | 339 (58.8%) | 2 | 72 (30.6%) |
| I want to feel safe around other people | 4 | 285 (35.1%) | 4 | 248 (43.0%) | 6 | 37 (15.7%) |
| I don't want to get really sick from COVID-19 | 5 | 265 (32.6%) | 5 | 227 (39.3%) | 5 | 38 (16.2%) |
| I believe life won't go back to normal until most people get a COVID-19 vaccine | 6 | 198 (24.4%) | 6 | 174 (30.2%) | 8 | 24 (10.2%) |
| I want to stop wearing masks | 7 | 159 (19.6%) | 7 | 139 (24.1%) | 9 | 20 (8.5%) |
| It is a requirement for my school or workplace | 8 | 115 (14.2%) | 8 | 106 (18.4%) | 11 | 9 (3.8%) |
| I have a chronic health problem, like asthma or diabetes | 9 | 103 (12.7%) | 9 | 89 (15.4%) | 10 | 14 (6.0%) |
| My doctor told me to get a COVID-19 vaccine | 10 | 88 (10.8%) | 10 | 80 (13.9%) | 12 | 8 (3.4%) |
| None | 11 | 67 (8.3%) | | | 4 | 67 (28.5%) |
| Other (unspecified) | 12 | 46 (5.7%) | 11 | 19 (3.3%) | 7 | 27 (11.5%) |
| Did not respond (missing) | | 19 (2.3%) | | 5 (0.9%) | | 14 (6.0%) |
| Selected any response (including 'Other' or 'None') | | 793 (97.7%) | | 572 (99.1%) | | 221 (94.0%) |
| Selected any response (besides 'None') | | 726 (89.4%) | | | | 154 (65.5%) |
| Selected any response (besides 'Other' or 'None') | | 692 (85.2%) | | 561 (97.2%) | | 131 (55.7%) |
| *Concerns* | | | | | | |
| I'm concerned about side effects from the vaccine | 1 | 258 (31.8%) | 2 | 168 (29.1%) | 1 | 90 (38.3%) |
| None | 2 | 178 (21.9%) | 1 | 178 (30.8%) | | |
| I don't trust that the vaccine will be safe | 3 | 125 (15.4%) | 4 | 61 (10.6%) | 2 | 64 (27.2%) |
| I don't know enough about how well a COVID-19 vaccine works | 4 | 119 (14.7%) | 3 | 77 (13.3%) | 4 | 42 (17.9%) |
| I'm not concerned about getting really sick from COVID-19 | 5 | 98 (12.1%) | 5 | 56 (9.7%) | 4 | 42 (17.9%) |
| I don't like needles | 6 | 95 (11.7%) | 5 | 56 (9.7%) | 6 | 39 (16.6%) |
| Other (unspecified) | 7 | 71 (8.7%) | 8 | 27 (4.7%) | 3 | 44 (18.7%) |
| I don't think vaccines work very well | 8 | 61 (7.5%) | 7 | 34 (5.9%) | 9 | 27 (11.5%) |
| I am concerned COVID-19 vaccines may cause infertility | 9 | 53 (6.5%) | 8 | 27 (4.7%) | 10 | 26 (11.1%) |
| I am concerned the COVID-19 vaccines contain fetal cells | 10 | 50 (6.2%) | 16 | 17 (2.9%) | 7 | 33 (14.0%) |
| I already had COVID-19 | 11 | 48 (5.9%) | 10 | 26 (4.5%) | 13 | 22 (9.4%) |
| I have religious reasons not to vaccinate | 12 | 47 (5.8%) | 17 | 14 (2.4%) | 7 | 33 (14.0%) |
| I am concerned I could get COVID-19 from the vaccine | 13 | 43 (5.3%) | 14 | 19 (3.3%) | 12 | 24 (10.2%) |
| I am concerned I may need to miss work if I feel sick from the COVID-19 vaccine | 14 | 42 (5.2%) | 11 | 25 (4.3%) | 15 | 17 (7.2%) |
| I don't believe the COVID-19 pandemic is as bad as some people say it is | 15 | 39 (4.8%) | 15 | 18 (3.1%) | 14 | 21 (8.9%) |

*(Continued)*

**Table 3.** (Continued)

| Reason for/against vaccination | Overall (N = 812) | | Initiated COVID-19 vaccine (N = 577) | | Did not initiate COVID-19 vaccine (N = 235) | |
|---|---|---|---|---|---|---|
| | Rank | n (%) | Rank | n (%) | Rank | n (%) |
| I am concerned COVID-19 vaccines could change your DNA | 15 | 39 (4.8%) | 18 | 13 (2.3%) | 10 | 26 (11.1%) |
| I'm allergic to vaccines | 17 | 36 (4.4%) | 12 | 23 (4.0%) | 16 | 13 (5.5%) |
| I don't want to pay for it | 18 | 32 (3.9%) | 13 | 22 (3.8%) | 17 | 10 (4.3%) |
| I don't know when or where to get a COVID-19 vaccine | 19 | 21 (2.6%) | 19 | 12 (2.1%) | 20 | 9 (3.8%) |
| I am concerned I will need to provide a social security number or government ID to get the COVID-19 vaccine | 20 | 19 (2.3%) | 20 | 9 (1.6%) | 17 | 10 (4.3%) |
| I cannot get the vaccine from a place that I trust | 21 | 18 (2.2%) | 21 | 8 (1.4%) | 17 | 10 (4.3%) |
| It is difficult to get to a vaccination site to get the COVID-19 vaccine | 22 | 15 (1.8%) | 22 | 7 (1.2%) | 21 | 8 (3.4%) |
| Did not respond (missing) | | 65 (8.0%) | | 53 (9.2%) | | 12 (5.1%) |
| Selected any response (including 'Other' or 'None') | | 747 (92.0%) | | 524 (90.8%) | | 223 (94.9%) |
| Selected any response (besides 'None') | | 569 (70.1%) | | 346 (60.0%) | | |
| Selected any response (besides 'Other' or 'None') | | 689 (84.9%) | | 505 (87.5%) | | 184 (78.3%) |

(n = 235), the most commonly reported concern was side effects of the vaccine (38.3%); 27.2% reported they did not trust that the vaccine will be safe, 17.9% reported they did not know enough about how well a COVID-19 vaccine works, and 17.9% reported they were not concerned about getting very sick from COVID-19. Eighteen other concerns were reported by at least 8 respondents without vaccine initiation or 15 respondents overall. Motivators and concerns stratified by community cohort and initiation status are presented in **S1A–S1F Table** in **S1 File**.

## Factors associated with vaccination

Results for bivariate and multivariable models are presented in **Table 4**. In bivariate logistic regression models, vaccine hesitancy (OR 0.40, 95% CI:0.29–0.54) and low (vs high) social

**Table 4. Factors associated with COVID-19 vaccine initiation, Colorado, 2021.**

| Characteristic | Bivariate | | | | | Multivariable | |
|---|---|---|---|---|---|---|---|
| | Total[a] (N = 812) | Initiated COVID-19 Vaccine[a] | | Unadjusted Models[b] | | Adjusted Model[b,c] | |
| | | Yes (N = 577) | No (N = 235) | OR (95% CI) | P value | OR (95% CI) | P value |
| **Age (years)** | 42.2 (16.8) | 44.5 (17.2) | 36.3 (14.1) | 1.03 (1.02, 1.05) | < .001 | 1.02 (1.01, 1.04) | 0.002 |
| **Gender** | | | | | | | |
| Man | 338 (42.1%) | 223 (39.1%) | 115 (49.4%) | 0.66 (0.43, 1.01) | 0.06 | 0.71 (0.43, 1.15) | 0.16 |
| Woman (ref.) | 465 (57.9%) | 347 (60.9%) | 118 (50.6%) | reference | | reference | |
| **Community cohort** | | | | | | | |
| Urban Latina/o/x | 189 (23.3%) | 140 (24.3%) | 49 (20.9%) | 1.01 (0.41, 2.47) | 0.98 | 0.70 (0.36, 1.38) | 0.30 |
| Rural Latina/o/x (ref.) | 195 (24.0%) | 144 (25.0%) | 51 (21.7%) | reference | | reference | |
| Urban Black/AA | 169 (20.8%) | 105 (18.2%) | 64 (27.2%) | 0.58 (0.18, 1.84) | 0.36 | 0.68 (0.23, 2.00) | 0.48 |
| Rural Black/AA | 195 (24.0%) | 128 (22.2%) | 67 (28.5%) | 0.68 (0.30, 1.55) | 0.35 | 0.41 (0.14, 1.21) | 0.11 |
| Urban AI/AN | 64 (7.9%) | 60 (10.4%) | 4 (1.7%) | 5.31 (2.15, 13.13) | < .001 | 4.61 (2.20, 9.68) | < .001 |
| **Immigrant (born outside the US)** | | | | | | | |
| Yes | 265 (33.4%) | 183 (32.4%) | 82 (35.7%) | 0.87 (0.45, 1.69) | 0.67 | | |

*(Continued)*

**Table 4.** (Continued)

| Characteristic | Bivariate | | | | | Multivariable | |
| --- | --- | --- | --- | --- | --- | --- | --- |
| | Total[a] (N = 812) | Initiated COVID-19 Vaccine[a] | | Unadjusted Models[b] | | Adjusted Model[b,c] | |
| | | Yes (N = 577) | No (N = 235) | OR (95% CI) | P value | OR (95% CI) | P value |
| No (ref.) | 529 (66.6%) | 381 (67.6%) | 148 (64.3%) | reference | | | |
| **Education[d]** | | | | | | | |
| High school/GED or less (ref.) | 493 (60.7%) | 323 (56.0%) | 170 (72.3%) | reference | | reference | |
| Associate's/technical degree | 116 (14.3%) | 83 (14.4%) | 33 (14.0%) | 1.32 (0.86, 2.04) | 0.20 | 0.71 (0.43, 1.19) | 0.19 |
| Bachelor's/graduate degree | 117 (14.4%) | 105 (18.2%) | 12 (5.1%) | 4.61 (2.28, 9.32) | < .001 | 2.14 (1.02, 4.52) | < .05 |
| Prefer not to answer/missing | 86 (10.6%) | 66 (11.4%) | 20 (8.5%) | 1.74 (1.19, 2.54) | 0.004 | 2.31 (1.08, 4.94) | 0.03 |
| **Household income[d]** | | | | | | | |
| Less than $25,000 (ref.) | 300 (36.9%) | 194 (33.6%) | 106 (45.1%) | reference | | reference | |
| $25,000-$49,999 | 193 (23.8%) | 139 (24.1%) | 54 (23.0%) | 1.41 (0.97, 2.03) | 0.07 | 0.86 (0.66, 1.13) | 0.28 |
| $50,000-$74,999 | 87 (10.7%) | 71 (12.3%) | 16 (6.8%) | 2.42 (1.41, 4.17) | 0.001 | 1.38 (0.68, 2.81) | 0.38 |
| $75,000 or more | 72 (8.9%) | 63 (10.9%) | 9 (3.8%) | 3.82 (1.61, 9.11) | 0.002 | 1.28 (0.52, 3.16) | 0.59 |
| Prefer not to answer/missing | 160 (19.7%) | 110 (19.1%) | 50 (21.3%) | 1.20 (0.78, 1.85) | 0.41 | 1.09 (0.66, 1.80) | 0.73 |
| **Has usual place for medical care** | | | | | | | |
| Yes (ref.) | 702 (86.8%) | 520 (90.6%) | 182 (77.4%) | reference | | reference | |
| No | 107 (13.2%) | 54 (9.4%) | 53 (22.6%) | 0.36 (0.17, 0.73) | 0.005 | 0.53 (0.27, 1.05) | 0.07 |
| **Health insurance coverage** | | | | | | | |
| None/don't know (status or type) | 163 (20.1%) | 94 (16.3%) | 69 (29.5%) | 0.28 (0.17, 0.47) | < .001 | 0.49 (0.32, 0.77) | 0.002 |
| Public | 384 (47.4%) | 264 (45.8%) | 120 (51.3%) | 0.45 (0.28, 0.75) | 0.002 | 0.70 (0.41, 1.17) | 0.17 |
| Private (ref.) | 263 (32.5%) | 218 (37.8%) | 45 (19.2%) | reference | | reference | |
| **Health literacy score[e]** | 0.8 (1.3) | 0.8 (1.3) | 0.7 (1.3) | 1.02 (0.87, 1.20) | 0.82 | | |
| **Vaccination hesitancy, total score[f]** | 3.3 (1.0) | 3.1 (0.9) | 3.9 (1.1) | 0.40 (0.29, 0.54) | < .001 | 0.41 (0.32, 0.53) | < .001 |
| **Medical mistrust, total score[g]** | 2.9 (0.8) | 2.8 (0.8) | 3.0 (0.9) | 0.78 (0.58, 1.04) | 0.09 | 1.20 (0.96, 1.50) | 0.11 |
| **COVID-19 vaccination social norms composite[h]** | | | | | | | |
| Low | 254 (31.6%) | 128 (22.3%) | 126 (55.0%) | 0.23 (0.13, 0.42) | < .001 | 0.48 (0.27, 0.84) | 0.01 |
| Medium | 213 (26.5%) | 172 (29.9%) | 41 (17.9%) | 0.95 (0.65, 1.37) | 0.77 | 1.07 (0.64, 1.78) | 0.80 |
| High (ref.) | 337 (41.9%) | 275 (47.8%) | 62 (27.1%) | reference | | reference | |

AA = African American; AI/AN = American Indian/Alaska Native; CI = confidence interval; OR = odds ratio; ref. = reference level; SD = standard deviation. Shaded cells indicate significance at the p < .05 level or less.

[a] Descriptive statistics are presented as either "n (%)" or "mean (SD)" and correspond to the observations included in bivariate models. Percentage denominators exclude missing data. [b] Modaels were specified as logistic regression with robust standard errors (generalized estimating equations with independence/variance components working correlation structure).

[c] Multivariable model excludes participants missing data for any retained variable. Total sample size for this model was N = 778, including N = 558 (71.7%) who initiated the COVID-19 vaccine and N = 220 (28.3%) who did not.

[d] Missing or "prefer not to answer" responses were treated as a category if representing at least 5% of the total sample.

[e] Range = 0 to 4; higher scores indicate more frequent need for help reading written information from doctor or drug store.

[f] Based on modified Vaccination Attitudes (VAX) Scale; range = 1 to 6; higher scores indicate greater vaccine hesitancy.

[g] Based on modified Group-Based Medical Mistrust Scale (GBMMS); range = 1 to 5; higher scores indicate greater group-based medical mistrust.

[h] Classified based on total score (range = 1 to 5) calculated as mean of 2 items (subjective and descriptive norms), with "low" = 1 to 2.5, "medium" = 3 to 3.5, "high" = 4 to 5. Higher scores reflect greater perceived social norms regarding vaccination (perceiving a larger percentage of people who would get or want the respondent to get a vaccine).

norms perceptions (OR 0.23, 95% CI:0.13, 0.42) were associated with lower odds of COVID-19 vaccination initiation. Participants from the AI/AN community had significantly higher odds of vaccine initiation (OR 5.31; 95% CI:2.15–13.13) compared with rural Latino/a/x. Additionally, increased age, higher education, and higher household income were all associated

with initiating COVID-19 vaccination. Not having a usual place for medical care and having public or no health insurance (compared with private insurance) were associated with lower odds of vaccination. Medical mistrust did not have a statistically significant association with vaccine initiation in bivariate analysis (OR 0.78, 95% CI:0.58–1.04, p = 0.09), but met the threshold for inclusion in the multivariable model (p<0.2). Health literacy and immigrant status were not associated with the primary outcome and were not included in the adjusted model.

After adjusting for other covariates, vaccine hesitancy (OR: 0.41 95% CI:0.32–0.53; p < .001) and low perceptions of social norms for COVID-19 vaccination (OR 0.48, 95% CI:0.27–0.84; p = .01) remained statistically significant predictors of COVID-19 vaccine initiation. The association of medical mistrust with the outcome of vaccine initiation remained non-significant. Increased age, higher education (completion of a bachelor's or graduate degree compared with high school or less), being uninsured, and being a member of the AI/AN community remained statistically significant predictors of initiation, while income, having public insurance, and having a usual source of medical care no longer had a statistically significant association with vaccination initiation.

## Discussion

### Vaccine initiation

This cross-sectional multiple cohort survey study explored COVID-19 vaccination in five Colorado communities that had been disproportionately impacted by the pandemic. Vaccine initiation rates among our sample are similar to rates found in other studies around the same timeframe [33, 34]. Specifically, the National Immunization Survey found cumulative initiation rates for COVID-19 vaccine were lower for Black/AA, Hispanic, and AI/AN populations in April 2021, but that disparities had lessened by November 2021, at the same time our data was collected [33]. However, while the national estimates for COVID-19 vaccine initiation for Black/AA (78.2%) and Hispanic (81.3%) adults were higher than those found in our Colorado sample, the national estimate for AI/AN (61.8%) adults was lower than in this Colorado study (94%). The National Immunization Survey also estimated vaccine coverage among the subgroup of Somali adults to be 52.6%. Our rural African American cohort primarily included Somali refugees resettled to Colorado and had a 65.6% initiation rate. While some of these differences may be due to differences in study design and sampling, our data suggest there may also be important state and regional variations in vaccine initiation as well as the underlying reasons for accepting the vaccines, which that warrant messaging tailored to local context and community needs [35].

### Vaccination concerns and motivators

Our sample's reported reasons for and against COVID-19 vaccines also aligned with other studies about COVID-19 vaccine intent. One national study found the most common concerns reported among non-white participants were about possible side effects and effectiveness of the vaccines [34]. This aligns with our top reported concerns; however, respondents to our survey also reported numerous concerns related to their understanding of how vaccines work and potential long-term health effects. The varied reported concerns speak to the need for COVID-19 vaccination efforts to clearly articulate the safety of the vaccines in both the short and long term and their ability to prevent infection and severe disease, including transparent discussion of real concerns about vaccines such as waning efficacy and rare but real safety concerns. Our findings also revealed that keeping oneself and loved ones safe were motivators, even among 30–40% of those who had not received a COVID-19 vaccine, suggesting there is

room to increase vaccine uptake and reduce disparities in communities where initiation has been low if people believe the vaccine will protect those they care about. Other vaccine promotion research has shown that personal values regarding safety (both in terms of protecting loved ones from both perceived risks of vaccines and harms of a communicable disease) and efficacy of vaccines in general are important factors influencing vaccination intention and completion [36].

## Factors in vaccine initiation

We also identified several factors associated with COVID-19 vaccine initiation in underserved Colorado communities. Notably, being uninsured or unknown insurance status predicted lower odds of vaccine initiation which suggests that, even though COVID-19 vaccines were free and distributed in community-based sites, there were still barriers to uptake for some. For example, in rural areas, access may have been limited by distance to mobile vaccine distribution sites. Some uninsured people may have worried about unanticipated out-of-pocket costs. While our findings align with some national survey findings highlighting racial disparities and the impact of socio-demographic factors, [37] they contradict others using the same dataset that found uninsurance status was not a predictor of hesitancy [38]. Another study found that over time, federally-qualified health centers delivered a greater proportion of the vaccines that 25.9 million of their patients received and nearly all racial groups achieved equity in vaccine initiation [39]. This suggests that delivery in a known and trusted healthcare setting (like an FQHC) may help improve access to and acceptance of COVID-19 vaccines for under- and uninsured patients [40].

## Hesitancy and mistrust factors in vaccine initiation

Additionally, we found that vaccine hesitancy, but not medical mistrust, was negatively associated with vaccine initiation. These findings are mixed in the context of other research around vaccine attitudes and trust. Another study of medical mistrust and COVID-19 trials and vaccines showed that higher levels of medical mistrust may predict refusal of COVID-19 vaccines [41, 42]. Our analyses showed that vaccine hesitancy, a construct more closely tied to trust in vaccines specifically, may be more important to initiation. Vaccine hesitancy is a global measure distinct from medical mistrust and our findings suggest that there may be something about vaccine attitudes that uniquely predicts initiation and may be a more appropriate focus of future work to improve COVID-19 vaccine uptake. However, this does not negate the need for the healthcare system to continue to earn the trust of communities that have experienced mistreatment in healthcare and research–rather it speaks to the need for this trust to be earned by addressing structural racism in healthcare and in specific areas of care like vaccination, as others have argued [40, 43, 44].

Finally, low perception of social norms of COVID-19 vaccination was a negative factor for COVID-19 vaccine initiation. Given the finding that the top motivators for vaccination were to keep family and community safe, it seems self-evident that perceiving others like oneself are not receiving the vaccine or expecting one to vaccinate would be a risk factor for not getting these vaccines. Rabb and colleagues found that social norms perceptions were the strongest predictor of vaccination [45] suggesting the need to encourage discussions between close relations about vaccination. Knowing this perception predicts vaccination behaviors may also open an opportunity for communities to promote vaccination among their members by sharing stories and developing collective expectations and increased social norms. amongst friends and family, suggesting that the.

## Strengths and limitations

This study has several limitations that influence how our findings should be interpreted. First and foremost, our study focused on five regional and racial/ethnic communities that experienced disparities in COVID-19 incidence and vaccination early in the pandemic. As such, our findings are not intended to represent the broad population's vaccination attitudes and behaviors. Similarly, our sample may not be large enough to generalize to broader populations and our for each of these focus communities. Additionally, the community-based venue and snowball sampling approaches utilized through our community connector model relied on networks within each community. Although our samples match the demographics of these communities, this approach may limit the generalizability of the findings and our ability to compare our findings to other studies in these communities. For example, participants may be somewhat more like each other than a random sample due to their social connections to each other; or those who were less hesitant about COVID-19 vaccination may have been more willing to participate in the study when recruited by the Community Connectors. As the vaccination uptake rates in our dataset were similar to national rates and those reported in other underserved communities at similar timepoints, it is possible the potential sampling bias resulting from our community-based data collection approach is minimal. Future applications of the community connector approach for data collection could be paired with feasible strategies for randomly selecting participants, which were not possible in our context, to lessen the potential biases in sampling. Additionally, the community connector model is a strength of the larger study in which this survey is embedded because it fosters relationship and familiarity between study participants and the research team for longitudinal follow-up across multiple waves of data collection [22–24]. This is intended to improve retention in communities often underrepresented in research. Our planned exploratory subgroup analysis was limited by reduced sample sizes for some communities, which prevented in-depth stratified analyses within each community. Limitations in resources and time constraints limited also constrained our ability to extend our recruitment timeline; however, our moderate to large sample sizes in some communities likely reduced sampling bias. We aim to address this in future waves of data collection through additional recruitment in some communities to achieve larger sample sizes.

To reduce survey burden, we also used adapted measures of two constructs–vaccine hesitancy and medical mistrust–which were shorter than the validated instruments. We consulted with community advisors in each focus population to ensure the brief measures we used were readable and captured key constructs from the original measure. While our intention in adapting the medical mistrust measure was to capture perceptions of the healthcare system as unequal and untrustworthy and not as a measure of distrust as a problem with community members, it likely remains an imperfect measure and may be difficult to compare to other studies on the role of mistrust in vaccination behavior. Nevertheless, this study of factors associated with COVID-19 vaccination initiation in five underserved communities in Colorado provides necessary data to inform vaccination promotion message development focused on these communities with lower vaccination uptake and documented health disparities and barriers to primary healthcare use. Our sample of 841, while moderate in size, may still inform interventions tailored to the needs of each community and our community-based data collection approach can inform additional data collection efforts to learn community-specific factors in COVID-19 vaccination.

## Implications for practice

Our findings suggest that varied approaches are necessary to increase vaccine uptake in communities that experience disparities in vaccination coverage. Healthcare providers and public

health officials need to be able to communicate effectively and persuasively with patients and communities about the safety and effectiveness of the COVID-19 vaccines at preventing severe disease and long COVID, as well as accurately describe the potential side-effects [46] and address concerns of patients [47]. These strategies may include interventions shown effective for other vaccines, such as motivational interviewing [48–53] and presumptive recommendations [50]. Other strategies that may be effective, especially in communities experiencing healthcare mistrust or other disparities in health literacy or healthcare access, include community engaged approaches such as Community Translation (sometimes called Boot Camp Translation) [35, 54, 55], which engages community members in translating medical evidence (such as COVID-19 vaccine safety) into relevant and responsive messages and dissemination plans tailored to their communities. These approaches can all be tailored to directly address the most common concerns or hindering factors associated with non-vaccination in a community and ground community education and messaging around the motivators and resources that support greater vaccination coverage.

## Conclusions

Our findings support the need for further interventions to increase vaccination, prevent COVID-19, and promote equity. These findings will inform further efforts to promote vaccination and build a trustworthy healthcare system and vaccine confidence through messages and initiatives that center family and community safety, vaccine safety and effectiveness, and address the social norms of vaccination against COVID-19 across–and in partnership with– communities. Future work will assess the impact of community co-developed messaging tailored to address these factors and designed to increase vaccination uptake in these communities.

## Supporting information

**S1 File.** S1a-S1f Table.
(PDF)

## Acknowledgments

The authors would like to extend our deepest gratitude to Charlene Barrientos Ortiz, our Community Connectors–Gloria Deloach, Mahad Dirieh, Lisa Lucero, and Stephanie Salazar-Rodriguez–and the many data collectors across these five Colorado communities who made this work possible. Without their dedication and hard work, we could not have completed this study.

## Author Contributions

**Conceptualization:** Sarah E. Brewer, Jenna E. Reno, Donald E. Nease.

**Data curation:** Kaitlyn B. Bertin, Krithika Suresh.

**Formal analysis:** Sarah E. Brewer, Kaitlyn B. Bertin, Krithika Suresh.

**Funding acquisition:** Sarah E. Brewer, Donald E. Nease.

**Investigation:** Sarah E. Brewer, Bethany M. Kwan, Donald E. Nease.

**Methodology:** Sarah E. Brewer, Krithika Suresh, Jenna E. Reno.

**Project administration:** Sarah E. Brewer, Kaitlyn B. Bertin, Crystal LoudHawk-Hedgepeth, Montelle Tamez, Jenna E. Reno.

**Supervision:** Sarah E. Brewer, Donald E. Nease.

**Validation:** Sarah E. Brewer.

**Writing – original draft:** Sarah E. Brewer.

**Writing – review & editing:** Kaitlyn B. Bertin, Krithika Suresh, Crystal LoudHawk-Hedgepeth, Montelle Tamez, Jenna E. Reno, Bethany M. Kwan, Donald E. Nease.

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
