## [Decision Letter · Decision Letter 0]

5 Mar 2024

PONE-D-24-00188Factors in COVID-19 Vaccine Uptake in Five Racial/Ethnic Colorado Communities: A report from the Colorado CEAL ProjectPLOS ONE

Dear Dr. Brewer,

Thank you for submitting your manuscript to PLOS ONE. After careful consideration, we feel that it has merit but does not fully meet PLOS ONE’s publication criteria as it currently stands. Therefore, we invite you to submit a revised version of the manuscript that addresses the points raised during the review process.

We look forward to receiving your revised manuscript.

Kind regards,

Wajdy Jum’ah Al-Awaida, Ph.D

Academic Editor

PLOS ONE

Journal Requirements:

 "Research reported in this publication was supported by the National Institutes of Health under the Sub-OTA No. 6793-02-S018. The content is solely the responsibility of the authors and does not necessarily represent the official views of the National Institutes of Health. 

Study data were collected and managed using REDCap electronic data capture tools hosted at the University of Colorado. REDCap (Research Electronic Data Capture) is a secure, web-based application designed to support data capture for research studies, providing: 1) an intuitive interface for validated data entry; 2) audit trails for tracking data manipulation and export procedures; 3) automated export procedures for seamless data downloads to common statistical packages; and 4) procedures for importing data from external sources. 

This project and related publication were supported by NIH/NCATS Colorado CTSA Grant Number UL1 TR002535. Its contents are the authors’ sole responsibility and do not necessarily represent official NIH views."

"This project and related publication were supported by NIH/NCATS Colorado CTSA Grant 

Number UL1 TR002535. Its contents are the authors’ sole responsibility and do not necessarily 

represent official NIH views.

Research reported in this publication was supported by the National Institutes of Health under 

the Sub-OTA No. 6793-02-S018. The content is solely the responsibility of the authors and does 

not necessarily represent the official views of the National Institutes of Health."

 "Research reported in this publication was supported by the National Institutes of Health under the Sub-OTA No. 6793-02-S018. The content is solely the responsibility of the authors and does not necessarily represent the official views of the National Institutes of Health. 

Study data were collected and managed using REDCap electronic data capture tools hosted at the University of Colorado. REDCap (Research Electronic Data Capture) is a secure, web-based application designed to support data capture for research studies, providing: 1) an intuitive interface for validated data entry; 2) audit trails for tracking data manipulation and export procedures; 3) automated export procedures for seamless data downloads to common statistical packages; and 4) procedures for importing data from external sources. 

This project and related publication were supported by NIH/NCATS Colorado CTSA Grant Number UL1 TR002535. Its contents are the authors’ sole responsibility and do not necessarily represent official NIH views."

"I have read the journal's policy and the authors of this manuscript have the following competing interests: Sarah Brewer served on an advisory board for Merck. The company did not play any role in this research and we do not receive research funding from them. All other authors have no conflicts to declare."

**Additional Editor Comments:**

Given the content and context of the manuscript titled "Factors in COVID-19 Vaccine Uptake in Five Racial/Ethnic Colorado Communities: A report from the Colorado CEAL Project," here are some constructive negative comments for review:

1. **Clarity and Structure**: The manuscript could benefit from clearer section headings and subheadings to enhance readability. The flow between sections appears disjointed, making it challenging for readers to follow the progression of the study.

2. **Data Analysis**: The methods section could provide more detailed information on the statistical analysis to enhance transparency. For instance, specifying the criteria for including variables in the multivariable model could clarify the analytical choices made.

3. **Sample Representativeness**: Concerns about the representativeness of the sample could be addressed more thoroughly. While the study targets specific racial/ethnic communities, it's unclear how these groups were selected and if they accurately represent the broader population's vaccine attitudes and behaviors.

4. **Limitations**: While some limitations are acknowledged, a more comprehensive discussion on how these limitations impact the study's findings and implications would strengthen the manuscript. For example, considering the potential biases introduced by the survey methodology or the impact of external factors on vaccine uptake during the study period.

5. **Implications for Practice**: The manuscript could more explicitly link its findings to practical applications. Suggestions for how healthcare providers, policymakers, and community leaders can use this information to improve vaccine uptake strategies would be valuable.

6. **Literature Review**: The introduction and discussion could be strengthened by incorporating a broader range of literature, especially recent studies that offer contrasting or complementary findings. This would help situate the study within the current research landscape more effectively.

These comments aim to provide constructive feedback to enhance the manuscript's clarity, depth, and contribution to the field.

Reviewers' comments:

Reviewer's Responses to Questions

**Comments to the Author**

1. Is the manuscript technically sound, and do the data support the conclusions?

Reviewer #1: Yes

Reviewer #2: Yes

Reviewer #3: Yes

2. Has the statistical analysis been performed appropriately and rigorously? 

Reviewer #1: Yes

Reviewer #2: No

Reviewer #3: Yes

3. Have the authors made all data underlying the findings in their manuscript fully available?

Reviewer #1: Yes

Reviewer #2: Yes

Reviewer #3: Yes

4. Is the manuscript presented in an intelligible fashion and written in standard English?

Reviewer #1: Yes

Reviewer #2: Yes

Reviewer #3: Yes

5. Review Comments to the Author

Reviewer #1: The only point that is concerning about the manuscript is the small sample size that were studied; 841 participants were involved in the study but 29 participants were excluded from the study, making the total participants 812. Hence, although the results are convincing, but I don’t believe that the 812 participants are enough to generalize the results that authors are presenting. Hence my recommendation, is to involve more participants (if possible), this would enhance the strength of the manuscript and its conclusions.

Reviewer #2: The manuscript presents a clear and structured investigation into the motivators, concerns, and factors associated with COVID-19 vaccine initiation across diverse communities in Colorado. After reviewing the manuscript "Factors in COVID-19 Vaccine Uptake in Five Racial/Ethnic Colorado Communities: A report from the Colorado CEAL Project," we have identified several areas that warrant further attention to strengthen the manuscript:

1. Your manuscript provides a clear overview of the study's objectives and findings. The use of community-based data collection and logistic regression models is well-articulated. To enhance clarity, consider providing more explicit connections between the literature review and the study's specific contributions to the field.

2. The community connector model for participant recruitment is innovative but may introduce selection bias, limiting the generalizability of the findings. A discussion on how this sampling approach might have influenced the results and how future studies could address this limitation would be beneficial.

3. The application of bivariate and multivariable logistic regression models is appropriate. However, a more detailed rationale for the selection of variables included in the models, along with any assumptions or limitations of the statistical methods used, would provide greater transparency and reproducibility.

4. Expanding the literature review to include recent studies on vaccine uptake in similar and diverse contexts could provide a more comprehensive background and highlight the study's unique contributions more effectively.

5. The community-based approach and the focus on specific racial/ethnic communities in Colorado are notable strengths of the study. Elaborating on how these aspects bring new insights into vaccine uptake research could underscore the manuscript's innovativeness.

6. The study's exemption by the Colorado Multiple Institutional Review Board (COMIRB) is noted. Including more details on the ethical considerations, especially in terms of participant consent and data protection, would enhance the manuscript's ethical rigor.

7. The manuscript effectively utilizes data to support its findings. Ensuring that all figures and tables are clearly labeled and directly support the narrative will further enhance the presentation and interpretation of the results.

8. The manuscript references relevant literature, but ensuring the inclusion of the most current and directly related studies will strengthen the background and support for the study's findings.

9. The reliance on community connectors for recruitment, while community-engaged, may limit the representativeness of the sample. Addressing how this approach may influence the findings and suggesting ways to mitigate such limitations in future research would be valuable.

10. The adaptation of measures from longer validated instruments is practical but may affect their reliability and validity. Discussing the implications of these adaptations on the study's conclusions would provide a more nuanced understanding of the findings.

11. The specific context and methods of the study may limit the generalizability of the findings. A discussion on the applicability of the results to other settings and populations would provide valuable insights for readers.

Addressing these points will enhance the manuscript's clarity, rigor, and contribution to the field, making it a valuable addition to the literature on COVID-19 vaccine uptake in diverse communities.

Reviewer #3: General Comments:

• The manuscript provides valuable insights into COVID-19 vaccine uptake among racially and ethnically diverse communities in Colorado. It highlights important factors influencing vaccination decisions, which are crucial for designing effective public health interventions.

• Incorporating comparisons and findings from Hatmal et al. (2022) and J. Al-Awaida et al. (2021) throughout the manuscript could enhance the discussion by providing additional perspectives on vaccine hesitancy, adverse effects, and the impact of COVID-19 variants.

• Future studies could benefit from exploring the role of digital tools and machine learning in predicting vaccine uptake and addressing hesitancy, as suggested by the methodologies employed in Hatmal et al. (2022).

Overall, the research offers valuable insights into community-level vaccine uptake dynamics and contributes to the broader understanding of public health interventions' effectiveness in promoting COVID-19 vaccination. However, there are several areas where the study could be strengthened:

1. Comparison with Existing Literature: The paper could benefit from a more thorough comparison with existing studies, specifically in discussing its findings in the context of reported adverse effects and attitudes following COVID-19 vaccination among different populations. References such as Hatmal et al. (2022) and Al-Awaida et al. (2021) provide relevant insights into adverse effects, attitudes, and the impact of SARS-CoV-2 variants which could enhance the discussion around vaccine hesitancy and social norms.

2. Limitations: The study acknowledges its limitations, including potential biases due to its sampling approach and the generalizability of the findings. It is commendable that the paper discusses these limitations; however, it could further explore the impact of these limitations on the study's findings and propose strategies for addressing them in future research.

3. Predictive Modeling: While the paper effectively uses logistic regression to identify factors associated with vaccine initiation, incorporating predictive modeling techniques, as discussed in Hatmal et al. (2022) using machine learning tools, could provide deeper insights into the likelihood of vaccine initiation based on predisposing factors. This approach could add a novel dimension to the study by offering predictive analytics to identify individuals at risk of vaccine hesitancy.

4. Cultural and Socio-economic Factors: The paper could expand its discussion on the role of cultural and socio-economic factors in vaccine hesitancy and uptake. Integrating findings from Al-Awaida et al. (2021), which examines the correlates of SARS-CoV-2 variants on health outcomes across different continents, could provide a more nuanced understanding of how these factors influence vaccine attitudes and behaviors in diverse communities.

In conclusion, the paper makes a significant contribution to understanding COVID-19 vaccine uptake in racially and ethnically diverse communities. By addressing these suggested areas for improvement, future iterations of this research could offer even more comprehensive insights into effective strategies for promoting vaccine uptake and addressing public health disparities.

references:

1. J. Al-Awaida, W., Jawabrah Al Hourani, B., Swedan, S., Nimer, R., Alzoughool, F., J. Al-Ameer, H., ... & R. Hadi, N. (2021). Correlates of SARS-CoV-2 Variants on Deaths, Case Incidence and Case Fatality Ratio among the Continents for the Period of 1 December 2020 to 15 March 2021. Genes, 12(7), 1061.

2. Hatmal, M. M. M., Al-Hatamleh, M. A., Olaimat, A. N., Mohamud, R., Fawaz, M., Kateeb, E. T., ... & Bindayna, K. M. (2022). Reported adverse effects and attitudes among Arab populations following COVID-19 vaccination: a large-scale multinational study implementing machine learning tools in predicting post-vaccination adverse effects based on predisposing factors. Vaccines, 10(3), 366.

6. PLOS authors have the option to publish the peer review history of their article (what does this mean?). If published, this will include your full peer review and any attached files.

Reviewer #1: No

Reviewer #2: No

Reviewer #3: No

---

## [Author Response · Author response to Decision Letter 0]

17 May 2024

We have provided a detailed table of responses to each comment in an attached document.

---

## [Editor Report · Decision Letter 1]

27 May 2024

Factors in COVID-19 Vaccine Uptake in Five Racial/Ethnic Colorado Communities: A report from the Colorado CEAL Project

PONE-D-24-00188R1

Dear Dr. Brewer,

We’re pleased to inform you that your manuscript has been judged scientifically suitable for publication and will be formally accepted for publication once it meets all outstanding technical requirements.

Kind regards,

Wajdy Jum’ah Al-Awaida, Ph.D

Academic Editor

PLOS ONE
---

## [Editor Report · Acceptance letter]

2 Jun 2024

PONE-D-24-00188R1 

PLOS ONE

Dear Dr. Brewer, 

I'm pleased to inform you that your manuscript has been deemed suitable for publication in PLOS ONE. Congratulations! Your manuscript is now being handed over to our production team.

Kind regards, 

on behalf of

Prof. Wajdy Jum’ah Al-Awaida 

Academic Editor

PLOS ONE